# Experimental Investigation and Micromechanical Modeling of Elastoplastic Damage Behavior of Sandstone

**DOI:** 10.3390/ma13153414

**Published:** 2020-08-03

**Authors:** Chaojun Jia, Qiang Zhang, Susheng Wang

**Affiliations:** 1School of Civil Engineering, Central South University, Changsha 410075, China; jiachaojun@csu.edu.cn; 2Key Laboratory of Ministry of Education for Geomechanics and Embankment Engineering, Hohai University, Nanjing 210098, China; 3Research Institute of Geotechnical Engineering, China Institute of Water Resources and Hydropower Research, Beijing 100048, China; zhangq@iwhr.com; 4State key laboratory of hydraulics and mountain river engineering, Sichuan University, Chengdu 610065, China; 5Laboratory State Key Laboratory for Geomechanics and Deep Underground Engineering, School of Mechanics and Civil Engineering, China University of Mining and Technology, Xuzhou 221116, China; 6Laboratory of Multiscale and Multiphysics Mechanics, University of Lille, CNRS FRE 2016, LaMcube, 59000 Lille, France

**Keywords:** sandstone, elastoplastic, damage mechanics, micromechanics, return mapping algorithm

## Abstract

The mechanical behavior of the sandstone at the dam site is important to the stability of the hydropower station to be built in Southwest China. A series of triaxial compression tests under different confining pressures were conducted in the laboratory. The critical stresses were determined and the relationship between the critical stress and confining pressure were analyzed. The Young’s modulus increases non-linearly with the confining pressure while the plastic strain increment $N_{\phi }$ and the dilation angle *ϕ* showed a negative response. Scanning electron microscope (SEM) tests showed that the failure of the sandstone under compression is a coupled process of crack growth and frictional sliding. Based on the experimental results, a coupled elastoplastic damage model was proposed within the irreversible thermodynamic framework. The plastic deformation and damage evolution were described by using the micromechanical homogenization method. The plastic flow is inherently driven by the damage evolution. Furthermore, a numerical integration algorithm was developed to simulate the coupled elastoplastic damage behavior of sandstone. The main inelastic properties of the sandstone were well captured. The model will be implemented into the finite element method (FEM) to estimate the excavation damaged zones (EDZs) which can provide a reference for the design and construction of such a huge hydropower project.

## 1. Introduction

Thanks to their good geological and mechanical properties, sandstones serve as the privileged candidate materials for many rock engineering applications, such as hydropower engineering, petroleum engineering, road engineering, and other engineering applications [1]. Some examples of projects involving excavation in sandstones include the Xiang Jiaba hydropower station in Southwest China and railway tunnels in Northwest China. In this context, attaining a deeper understanding of the mechanical properties of sandstone is of crucial importance to the design and construction of rock engineering projects in such host rock.

The uniaxial and triaxial tests with cylindrical specimens are the most useful approach to study the mechanical properties of sandstone. Wong et al. [2] conducted triaxial compression tests on six sandstones with porosities ranging from 15% to 35% to investigate the deformation behavior of porous sandstones. The results showed that the sandstones experience from brittle faulting to cataclastic flow with the increase of effective pressure. Additionally, the tests indicated that a more porous and coarse-grained sandstone tends to be less brittle. Wasantha et al. [3] performed uniaxial compression tests on three sandstones with different grain sizes to study the strain rate on the mechanical behavior of sandstone. The observation suggested that the fine-grained sandstone is more responsive to strain rate compared to coarse-grained sandstone. Heap et al. [4] reported the influence of temperature on the short-term and time-dependent strengths of three sandstones under triaxial compression. A systematic reduction in strength during short term tests and an increase by several orders of magnitude during creep tests were observed in all three kinds of sandstone. The inherently anisotropic properties of sandstones were investigated [5,6]. The results showed that the permeability, Young’s modulus, compression strength, and tensility are all affected by the structural anisotropy. The influences of water-weakening, chemical corrosion, and heat treatment effects on the mechanical behavior of sandstone were investigated [7,8,9]. From the above studies, we conclude that the mechanical behavior of sandstone is significantly influenced by the grain size, porosity, and inherent structure which means different types of sandstone have a different response to stress and geological environments. Although the mechanical behavior of sandstone under uniaxial and triaxial conditions has been studied systematically in the literature, the mechanical properties of the sandstone from the specific dam site still should be investigated to provide useful data for the design and construction of the hydropower station [10].

Based on a large number of experimental results, the sandstones are assumed to belong to brittle geomaterials [11]. The mechanical behavior of such geomaterials exhibits elastic degradation, material hardening/softening, irreversible deformation, volumetric dilatancy, stress-induced anisotropy, etc. All these features of sandstones make it difficult to develop a constitutive model to capture the behavior of the brittle material. The purely plastic models cannot reveal the real dissipation mechanism of failure because the damage induced by the nucleation and propagation of microcracks is not considered [12,13]. A large number of elastic damage models following the continuum damage mechanics (CDM) are proposed. However, they cannot simulate the accumulation of irreversible plastic strain [14]. It is widely accepted that the damage caused by the microcracking, as well as the frictional sliding along the crack surfaces, are two mainly dissipative mechanisms governing stiffness deterioration, irreversible deformation, and progressive failure of brittle rock [15,16]. Besides, these two dissipation processes are inherently coupled with each other during the failure process. Therefore, the coupled elastoplastic damage models are the most appropriate approach to brittle rock such as sandstone.

In the past few decades, substantial progress has been made in modeling coupled elastoplastic deformation and induced damage of brittle rocks using continuum damage mechanics (CDM) [17,18,19]. These phenomenological models are formulated within the framework of irreversible thermodynamics, and they are easily implemented and applied to real engineering situations [20]. During these models, the total free energy, acting as the thermodynamic potential, can be expressed as a function of internal variables (plastic strain, damage variable, plastic hardening variable, etc.). In this context, the thermodynamic potential is decomposed into elastic free energy and locked plastic energy [21,22]. Both damage evolution and plastic flow depend on the formation of the locked plastic energy. However, the assumption of locked plastic energy has never been theoretically and experimentally justified. In order to describe the complex mechanical behaviors of rock materials, many model parameters are introduced but have no physical significance. In addition, such parameters are rarely determined from laboratory tests.

On the other hand, the micromechanical approaches provide an alternative way to deal with coupled plastic damage problems. These models consider the growth and frictional sliding of microcracks at relevant scales and determine the mechanical properties of cracked materials by an up-scaling method [23,24,25,26,27]. These models contain generally far fewer parameters compared to phenomenological ones. However, most of the micromechanical models were not formulated within the framework of irreversible thermodynamics. For practical rock mechanics problems such as the dam site sandstone, the predictive capability of these micromechanical based models is still under discussion [28].

In this study, the mechanical properties of sandstone collected from a dam foundation were systematically investigated. The basic inelastic mechanical behavior of the sandstone will be outlined. Based on the experimental results, a comprehensive analysis was performed on building a coupled elastoplastic damage model within the framework of irreversible thermodynamics. The damage evolution and plastic flow rules were developed according to the micromechanical based homogenization method. A new computational integration algorithm was proposed to deal with the coupled elastoplastic damage model. After the identification of the model’s parameters, the proposed model was applied to simulate the experimental results of the sandstone under uniaxial/triaxial compression conditions. The proposed model would help with the design and construction of a huge hydropower project using sandstone.

## 2. Experimental Investigations

### 2.1. Description of Sandstone

The sandstone used throughout this study was collected from the dam site of a hydropower project in Southwest China. The X-ray diffraction (XRD) and optical microscopy (OM) tests suggested that the mineralogical composition of this sandstone is about quartz (55%), feldspar (25%), sandy and clay detritus (20%). The quartz is of monocrystals of about 0.1–1.0 mm in size while the feldspar has granular structural and large phenocryst crystals. The mean density and initial porosity are 2.23 g/cm^3^ and 8.43%, respectively. A thin section of sandstone was prepared for the SEM test and the result is given in Figure 1. From the 2000× enlarged image, we can see that the sandstone is a blocky structure. The grains of sandstone are tightly cemented together. Many pores and microcracks are uniformly distributed. Besides, most of the sizes of the initial defects are less than 5 μm.

Because of its significant impact on the safety of the dam foundation, the permeability evolution of this sandstone in the triaxial and hydrostatic compression has been studied sufficiently [29,30]. The change of permeability with crack growth under different pore pressures was studied. The inert gas test technique was developed to measure the permeability of this sandstone. This study is devoted to the experimental and numerical investigations on the mechanical properties of this sandstone. All the cylindrical specimens were drilled and polished from the same block of material to a diameter of 50 mm and approximately 100 mm in length. The tolerance of straightness and flatness of the samples meets the requirement of the International Society for Rock Mechanics and Rock Engineering (ISRM) suggested method [31].

### 2.2. Experimental Method

A servo-controlled rock mechanics experimental system was used to complete all the experiments (Figure 2). This apparatus comprises a triaxial cell, three high-pressure servo-controlled pumps, and a data monitoring system. The confining pressure and pore pressure up to 60 MPa are loaded through separated pumps. The maximum deviatoric stress is 375 MPa. The axial strain $\varepsilon_{1}$ is measured by two linear variable displacement transducers (LVDT) with a resolution of ±1 μm, while the radial deformation $\varepsilon_{3}$ is monitored continuously using a ring radial displacement transducer chain wrapped tightly around the middle height of the specimen. All the stress and strain data are monitored and recorded continuously by an integrated acquisition system.

Following the ISRM suggested method, the conventional triaxial compression tests are performed in two steps. The confining pressure is loaded to the desired value. Then, the deviatoric stress $\left(\sigma_{1}-\sigma_{3}\right)$ is increased in the displacement control method until specimen failure while the confining pressure is kept constant during the whole process.

### 2.3. Experimental Results and Discussions

The conventional uniaxial/triaxial compression test results are presented in Figure 3. Consistently, the stress–strain curves of sandstone under different confining pressures can be divided into four symptomatic stages. (1) The curves are concave upward, a consequence of the closure of pre-existing defects. With the increase of confining pressure, the initial hardening is usually not marked because the pre-existing defects have already closed before the additional stress is applied. (2) The curves are approximately linear, as the elastic deformation of the grains is in the dominant role. (3) The curves depart from perfectly elastic behavior as a large number of microcracks proliferate and propagate throughout the specimens. (4) The failure of the specimens marks the significant drop of the curves due to the macroscopic fractures developed by the lining-up of microcracks. Furthermore, the frictional sliding capacity of fractures finally sustains the relative flattening of the curves.

Based on the recorded axial strain $\varepsilon_{1}$ and radial strain $\varepsilon_{3}$, the volumetric strain can be calculated according to the relation: $\varepsilon_{v}=\varepsilon_{1}+2\varepsilon_{3}$. The evolution of the volumetric strain during a typical experimental is presented in Figure 4. Additionally, a virtual elastic reference line is added based on the extrapolation of the linear part of the volumetric strain. According to the figure, the volumetric strain switches from the compaction-dominated phase to the dilatancy-dominated phase. Three typical points map the evolution process of the volumetric strain. The first is the onset of dilatancy $C^{\prime }$, which can be determined at the point where the volumetric strain departs from the approximately linear part [32]. This implies that at stress beyond the $C^{\prime }$ the deviatoric stress induces the pore structure to dilate. The point CD is a reversal point where the volumetric strain changes from compaction to dilatancy. Point D represents the volume of the specimen returning to the initial value.

The critical stresses at the three points and the corresponding peak stresses under different confining pressures are listed in Table 1. The results show a strong positive influence of confining pressure on the stresses. The influence of $\sigma_{3}$ on the corresponding stresses is given in Figure 5. The linear Mohr–Coulomb criterion and the nonlinear Hoek–Brown criterion are adapted to fit the experimental results:(1)$$\sigma_{1}-\frac{1+\sin\varphi }{1-\sin\varphi }\sigma_{3}-\frac{2c\cos\varphi }{1-\sin\varphi }=0$$
(2)$$\sigma_{1}-\sigma_{3}-\mathrm{UCS}{\left(m\frac{\sigma_{3}}{\mathrm{UCS}}+s\right)}^{a}=0$$
where *c* is the cohesion and φ is the frictional angle, UCS is the uniaxial compressive stress, *m*, *s* and *a* are all material constants. For the intact rock, the parameters *s* and *a* are fixed to 1.0 and 0.5, respectively. The fitting results are plotted in Figure 5 and the obtained strength parameters are presented in Table 2.

In general, the Mohr–Coulomb provides a better representation of the experimental data. The determined cohesion *c* and frictional angle φ all increase during the hardening process. Even though the fitting *R^2^s* are all high than 0.97, the determined USC at the peak point (25.295 MPa) is much less than the experimental data.

The $E_{s}$ is identified using the local gradient of a third-order polynomial fitted to the axial stress–strain curve and is given by $E_{s}=\partial \left(\sigma_{1}-\sigma_{3}\right)/\partial \varepsilon_{1}$ Similarly, the Poisson’s ratio is calculated by $\upsilon =-\partial \varepsilon_{3}/\partial \varepsilon_{1}$ [33]. Figure 6 illustrates the influence of the confining pressure on the deformation parameters ($E_{s}$ and  υ) of the sandstone. The Young’s modulus increases non-linearly with the confining pressure. When the confining pressure is lower, the $E_{s}$ increases relatively fast. The evolution of the Poisson’s ratio within the test confining pressure is not clear. The mean value of the Poisson’s ratio is 0.198.

As determined above, dilatancy, which is characterized by the dilation angle ϕ, is a significant property of the sandstone. To assess the dilation angle from uniaxial or triaxial tests, Vermeer and De Borst [34] proposed the equation
(3)$$\phi =\arcsin\frac{d\varepsilon_{v}^{p}}{-2d\varepsilon_{1}^{p}+d\varepsilon_{v}^{p}}$$

In Equation (3), $d\varepsilon_{1}^{p}$ and $d\varepsilon_{v}^{p}$ are axial and volumetric plastic strain increments, respectively. For ϕ>0, the irreversible radial strain increment is larger than that of the axial strain, which indicates that the plastic volumetric strain increases. While ϕ<0, the plastic contraction occurs [35]. The dilation angle is approved to relate to the plastic strain and the plastic strain increment $N_{\phi }$ is always introduced
(4)$$N_{\phi }=\frac{1+\sin\phi }{1-\sin\phi }=\tan^{2}\left(45+\phi /2\right)$$

The relationship between dilation parameters ($N_{\phi }$, *ϕ*) and confining pressure is presented in Figure 7. Both the plastic strain increment $N_{\phi }$ and the dilation angle *ϕ* show a negative response to the confining pressure. Also, the drop of $N_{\phi }$ is more significant under the lower $\sigma_{3}$. An exponential function is introduced to approach the correlation between $N_{\phi }$ and $\sigma_{3}$. The correlation coefficient $R^{2}$ is 0.999.

The stress–strain curves during the cyclic loading tests under the confining pressure of 10 MPa is shown in Figure 8 together with the data from the monotonic compression test. Three key observations arising from the similarity of the envelope curves of the tests during different loading histories. Firstly, the peak stress and residual stress of the sandstone are independent of how the specimen is loaded. Secondly, the axial and radial envelope curves during the cyclic loading coincide well with the monotonic loading before peak stress. Thirdly, the radial strain response in the post-peak region is quite larger than that of the monotonic loading due to the strain localization and the location and orientation of stress-induced fractures [36].

The SEM tests were investigated on the thin sections prepared from the fractured surfaces of the failed cylindrical specimen. With the test results shown in Figure 9, the micromechanics of damage is investigated. Microcracks with a width of approximately 1–2.5 μm are observed. Enlarging the microcracks to 5000×, we can see that the surfaces of the microcracks are relatively smooth while apparent dislocation can reach 5 μm. This irreversible deformation is originated from the growth of the cracks.

## 3. Thermodynamic Framework

Based on the mechanical tests and microscopic observation, it has been well confirmed that the failure of brittle sandstone can be attributed to the coupling between the irreversible deformation and damage induced by microcracks. The basic physical process of damage is the initiation, propagation, and coalescence of microcracks. In the compression stress state, the frictional sliding along the rough surfaces of the microcracks produces the irreversible deformation. In this context, a coupled elastoplastic damage model is more adequate to reproduce the inherent coupling between the plastic and damage dissipation processes. For the brittle rock materials, the assumption of small strains is appropriate. In the isothermal conditions, the total strain tensor **ε** can be decomposed into an elastic strain $\varepsilon^{e}$ and a plastic strain $\varepsilon^{p}$ according to the plastic theory
(5)$$\varepsilon =\varepsilon^{e}+\varepsilon^{p},\;\mathrm{and}\;d\varepsilon =d\varepsilon^{e}+d\varepsilon^{p}$$

The propagation of microcracks in rock materials is generally with faces parallel to the applied stress which results in stress-induced anisotropic damage of materials [37]. In this paper, we emphasize the formulation of a coupled elastoplastic damage model. For the sake of simplicity, an isotropic damage variable is adopted in this work. Therefore, an internal scalar variable ω is introduced to describe the growth of microcracks. Assuming an isothermal process and small strains, the free energy Ψ, taken as the thermodynamic potential, can be expressed in terms of a set of state variables: elastic strain $\varepsilon^{e}$, internal plastic variable *κ*, and damage variable ω
(6)$$\psi =\psi \left(\varepsilon^{e},\kappa ,\omega \right)$$

For any dissipative process, the Clausius–Duhem inequality must be satisfied [38]
(7)σ:dε−dψ≥0
where **σ** is the stress tensor. Substituting the differential of **ε** and Ψ into inequality (7), one gets
(8)$$\left(\sigma -\frac{\partial \psi }{\partial \varepsilon^{e}}\right):d\varepsilon +\frac{\partial \psi }{\partial \varepsilon^{e}}:d\varepsilon^{p}-\frac{\partial \psi }{\partial \kappa }d\kappa -\frac{\partial \psi }{\partial \omega }d\omega \geq 0$$

This relation should be satisfied for any values of state variables ($\varepsilon^{e}$, κ, and *ω*), and hence we have the state equation
(9)$$\sigma =\frac{\partial \psi }{\partial \varepsilon^{e}}$$

By defining the thermodynamic forces associated with the plasticity (*K*) and damage (*Y*), we have
(10)$$K=-\frac{\partial \psi }{\partial \kappa }$$
(11)$$Y=-\frac{\partial \psi }{\partial \omega }$$

To describe the plastic flow, a plastic potential $g^{p}=g^{p}\left(\sigma ,K,\omega \right)$ should be employed. Besides, a damage potential $g^{\omega }=g^{\omega }\left(Y,\omega \right)$ is introduced to describe the damage evolution. Finally, the rate of change of the internal variables can be characterized as
(12)$$d\varepsilon^{p}=d\lambda^{p}\frac{\partial g^{p}}{\partial \sigma }$$
(13)$$d\kappa =d\lambda^{p}\frac{\partial g^{p}}{\partial K}$$
(14)$$d\omega =d\lambda^{\omega }\frac{\partial g^{\omega }}{\partial Y}$$
where $d\lambda^{p}$ and $d\lambda^{\omega }$ are the plastic and damage multipliers, respectively. In the general case, a plastic criterion $f^{p}=f^{p}\left(\sigma ,K,\omega \right)$ is necessary to account for the pressure dependence of the brittle rock. Also, a damage criterion $f^{\omega }=f^{\omega }\left(Y,\omega \right)$ is introduced to describe the damage initiation. Therefore, the loading–unloading conditions can be represented by the Kuhn–Tucker conditions with the formulations
(15)$$f^{p}\left(\sigma ,K,\omega \right)\leq 0,d\lambda^{p}\geq 0,\;\mathrm{and}\;f^{p}\left(\sigma ,K,\omega \right)d\lambda^{p}=0$$
(16)$$f^{\omega }\left(Y,\omega \right)\leq 0,d\lambda^{\omega }\geq 0,\;\mathrm{and}\;f^{\omega }\left(Y,\omega \right)d\lambda^{\omega }=0$$

The first inequality in both Equations (15) and (16) suggests that the thermodynamic forces are within or on the yield surface and the second one indicates that the multipliers are nonnegative. The third equation ensures that the stress states lie on the yield surface during the complete loading or unloading process [13]. The two consistency conditions can be expressed as
(17)$$df^{p}=\frac{\partial f^{p}}{\partial \sigma }:d\sigma +\frac{\partial f^{p}}{\partial K}dK+\frac{\partial f^{p}}{\partial \omega }d\omega =0$$
(18)$$df^{\omega }=\frac{\partial f^{\omega }}{\partial Y}dY+\frac{\partial f^{\omega }}{\partial \omega }d\omega =0$$

In the general case $\left(f^{p}>0\;\mathrm{and}\;f^{\omega }>0\right)$, plastic flow and damage evolution take place in a coupled process. If $f^{p}>0$ and $f^{\omega }\leq 0$, only plastic flow occurs. If $f^{p}\leq 0$ and $f^{\omega }>0$, the material is in the elastic damage loading condition.

## 4. Microscopic Prediction

Under the general loading conditions (plastic flow is coupled with damage evolution), if the yield criteria ($f^{p}$, $f^{\omega }$) and potential functions ($g^{p}$, $g^{\omega }$) are given, the plastic and damage multipliers can be determined by solving the two consistency conditions Equations (15) and (16). Under a thermodynamic framework, the description of the damage conjugate force *Y* and plastic hard function *K* are related to the formation of thermodynamic potential (6). However, the assumption of thermodynamic potential when considering the coupled relationship between plasticity and damage has never been theoretically and experimentally justified in the phenomenological constitutive model. Moreover, many parameters are introduced, but have no physical significance. The micromechanical approaches provide an alternative way to deal with the coupled elastoplastic damage problems.

### 4.1. Effective Elastic Properties of Cracked Materials

The essential of micromechanical damage is to determine the macroscopic properties of materials from its microstructure (geometry, number, size, and spatial distribution of defects as well as their evolution laws) via homogenization schemes [39]. In this study, we consider a representative element volume (REV) Ω (with boundary ∂Ω) of a brittle material composed of an isotropic linear solid matrix and sets of defects in different orientations. According to the shapes, the defects in the solid matrix are assumed to be classified into two categories: penny-shaped microcracks and quasi-spherical pores. Then the ensemble of the solid matrix and pore therein is considered as a homogenized porous matrix with stiffness tensor $D^{m}$. Microcracks are classified into *N* families with the stiffness tensor $D^{c,r},\;r=1,2\ldots N$ according to the direction. According to the micromechanical method, both pore-weakened matrix and microcracks are considered as local elastic medium. The effective (homogenized) elasticity tensor of the microcrack–matrix rock system $D^{hom}$ is obtained by taking the average of the local strain over Ω [28,40]
(19)$$D^{\mathrm{hom}}=D^{m}+\sum_{r=1}^{N}\varphi_{r}\left(D^{c,r}-D^{m}\right):A^{c,r}$$
where $\varphi_{r}$ and $A^{c,r}$ are the volume fraction and average strain concentration tensor, respectively. The local strain ϵ is linear with the macroscopic one ε on ∂Ω, i.e., $\epsilon =A^{c}:\varepsilon$. is The stiffness tensor $D^{m}$ of rock matrix can be expressed as
(20)$$D^{m}=2\mu^{m}K+3k^{m}J$$
where $k^{m}$ is the bulk modulus and $\mu^{m}$ denotes shear modulus of rock materials. **J** and **K** are fourth order symmetric tensors
(21)$$J=\frac{1}{3}\delta ⊗\delta ,\;K=I-J$$
where **δ** is a second order unit tensor, and $I_{ijkl}=\frac{1}{2}\left(\delta_{ik}\delta_{jl}+\delta_{il}\delta_{jk}\right)$ is a fourth order unit symmetric tensor.

The sets of microcracks are considered as flat ellipsoids with radius $a_{r}$ and aspect ratio $\vartheta_{r}=c_{r}/a_{r}$ (Figure 10) where the subscript "*r*" stands for the *r*th family. The parameter *c* is the half-length of the small axis. The volume fraction $\varphi_{r}$ of the *r*th family can be expressed mathematically in the form
(22)$$\varphi_{r}=\frac{4}{3}\pi a_{r}^{2}c_{r}\frac{N}{\Omega }=\frac{4}{3}\pi \vartheta_{r}d_{r}$$
where $d_{r}=\frac{N}{\Omega }a_{r}^{3}$ is the well-known crack density parameter of the *r*th family and can be treated as an internal damage variable [41]. If the macroscopic damage ω is defined as the degradation of the elastic modulus, the microscopic damage variable *d* is certainly related to ω as ω=ω(d). In this context, the goal to determinate the homogenized elasticity tensor $D^{hom}$ is to evaluate the fourth-order concentration tensor $A^{c,r}$ for each phase. Several homogenization schemes can be found in the literature to deal with the concentration tensor. Considering that the Mori-Tanaka is especially suitable for a microcrack-matrix rock system [42], the homogenization procedure is based on this scheme and the determined homogenized elasticity tensor can be expressed in the form [43]
(23)$$D^{\mathrm{hom}}=\frac{1}{1+\eta_{1}d}3k^{m}J+\frac{1}{1+\eta_{2}d}2\mu^{m}K$$
where $\eta_{1}$ and $\eta_{2}$ are parameters only related to the Poisson’s ratio $\nu^{m}$ of a matrix, namely $\eta_{1}=\frac{16}{9}\frac{1-{\left(v^{m}\right)}^{2}}{1-2v^{m}}$ and $\eta_{2}=\frac{32}{45}\frac{\left(1-v^{m}\right)\left(5-v^{m}\right)}{2-v^{m}}$. Finally, the free energy of the matrix-cracks system can be expressed as
(24)$$\psi =\frac{1}{2}\left(\varepsilon -\varepsilon^{p}\right):D^{\mathrm{hom}}:\left(\varepsilon -\varepsilon^{p}\right)$$

Besides, the free energy (24) can also be expressed in the general form [27]
(25)$$\psi =\frac{1}{2}\left(\varepsilon -\varepsilon^{p}\right):D^{m}:\left(\varepsilon -\varepsilon^{p}\right)+\frac{1}{2}\varepsilon^{p}:D^{b}:\varepsilon^{p}$$
where the tensor $D^{b}$ can be expressed in the form
(26)$$D^{b}=\frac{1}{\eta_{1}d}3k^{m}J+\frac{1}{\eta_{2}d}2\mu^{m}K$$

Compared with the phenomenological model [44], a specific form of the locked plastic energy $\psi^{p}=\frac{1}{2}\varepsilon^{p}:D^{b}:\varepsilon^{p}$ can be determined with the micromechanical homogenization procedure. Besides, each parameter has clear physical meaning and can be determined from laboratory experiments.

### 4.2. Plastic Characterization

Based on the thermodynamic framework, the state Equation (9) can be expressed as
(27)$$\sigma =\frac{\partial \psi }{\partial \varepsilon^{e}}=D^{m}:\left(\varepsilon -\varepsilon^{p}\right)$$

In the microcrack-matrix rock system, the plastic strain $\varepsilon^{p}$ induced by the friction sliding along the crack surfaces is directly selected as the internal plastic variable κ. The thermodynamic force associated with the plastic strain $\varepsilon^{p}$ is divided according to Equation (10)
(28)$$\sigma^{p}=-\frac{\partial \psi }{\partial \varepsilon^{p}}=\sigma -D^{b}:\varepsilon^{p}$$
It shows that the thermodynamic force $\sigma^{p}$ is part of the macroscopic stress **σ**. The part $D^{b}:\varepsilon^{p}$ in the right hand of Equation (28) is the self-equilibrated stress in the solid matrix according to the homogenization scheme. Therefore, the thermodynamic force $\sigma^{p}$ is the local stress act on the closed microcracks.

Before formulating the plastic criterion, the plastic strain induced by the microcracks is decomposed into spherical strain tensor and deviatoric strain tensor. If the scalar variable β and vector Γ denote the shear dilation and friction sliding, the plastic strain is expressed as
(29)$$\varepsilon^{p}=\Gamma +\frac{1}{3}\beta \delta ,\;\beta =tr\varepsilon^{p}$$

The local stress $\sigma^{p}$ is also convenient to be decomposed into a deviatoric part and a spherical part
(30)$$\sigma^{p}=K:\sigma^{p}+J:\sigma^{p}=s^{p}+\sigma_{m}^{p}\delta$$
where $s^{p}=\sigma^{p}:K$ and $\sigma_{m}^{p}=\mathrm{tr}\sigma^{p}/3$ are deviatoric part and spherical part of the local stress.

According to the Equation (28), the local stress $\sigma^{p}$ can also be expressed with the form of macroscopic stress **σ**
(31)$$s^{p}=s-2\frac{1}{\eta_{2}d}\mu^{m}\Gamma ,\;\sigma_{m}^{p}=\sigma_{m}-\frac{1}{\eta_{1}d}k^{m}\beta$$
where $s=\sigma -\frac{1}{3}tr\sigma \delta$ and $\sigma_{m}=\frac{1}{3}tr\sigma$ are the deviatoric part and spherical part of the macroscopic stress, respectively.

To describe the local frictional sliding along the surfaces of microcracks, a generalized Coulomb criterion [45] is adopted in terms of local stress $\sigma^{p}$
(32)$$f^{p}=\left\|s^{p}\right\|+\eta_{\varphi }\sigma_{m}^{p}\leq 0$$
where $\eta_{\varphi }$ is the generalized friction coefficient due to the roughness of the crack surfaces. Equation (32) can also be rewritten in the form of macroscopic global stress **σ** in Equation (31)
(33)$$f^{p}=\left\|s-2\frac{1}{\eta_{2}d}\mu^{m}\Gamma \right\|+\eta_{\varphi }\left(\sigma_{m}-\frac{1}{\eta_{1}d}k^{m}\beta \right)$$

Under triaxial compression condition, the damage *d* at the peak point reaches $d_{c}$. The relationship between maximum principal stress $\sigma_{1}$ and minimum principal stress can be determined from (33) (under the sign convention in geomechanics)
(34)$$f^{p}=\sigma_{1}-\frac{2\eta_{\varphi }+\sqrt{6}}{\sqrt{6}-\eta_{\varphi }}\sigma_{3}-\frac{6}{\sqrt{6}-\eta_{\varphi }}\sqrt{\chi R\left(d_{c}\right)}=0$$
where *χ* is a constant which can be written with the parameters related to the Yong’s modulus and Poisson’s modulus [43]
(35)$$\chi =\frac{k^{m}\eta_{\varphi }^{2}}{2\eta_{1}}+\frac{\mu^{m}}{\eta_{2}}$$

In geomaterials, the micromechanical approach always introduces the non-associated plastic potential to describe the volumetric dilatancy during the failure process. However, the micromechanical analysis which uses the back stress term $D^{b}:\varepsilon^{p}$ to realize the hardening/softening behavior gives an alternative method [46]. Therefore, an associated flow rule $g^{p}=f^{p}$ is utilized to describe the evolution of plastic strain. According to the plastic theory (12), the rate of the plastic strain can be expressed as
(36)$$d\varepsilon^{p}=d\lambda^{p}\frac{\partial g^{p}}{\partial \sigma }=d\lambda^{p}\left(V+\frac{1}{3}\eta_{\varphi }\delta \right)$$
where $V=\frac{s^{p}}{||s^{p}||}$ is the flow direction. Compare Equation (36) with the rating form of plastic strain in Equation (29), one gets
(37)$$d\Gamma =d\lambda^{p}V,\;d\beta =d\lambda^{p}\eta_{\varphi }$$

The current value of plastic strain can be expressed in terms of the cumulated values of the plastic multiplier
(38)$$\varepsilon^{p}=\Lambda^{p}\left(V+\frac{1}{3}\eta_{\varphi }\delta \right),\;\Lambda^{p}=\int d\lambda^{p}$$

### 4.3. Damage Characterization

The damage criterion is a function of the damage conjugate force *Y* within the framework of irreversible thermodynamics. According to the Equation (11), the damage conjugate force *Y* can be determined with the function of free energy ψ.
(39)$$Y=-\frac{\partial \psi }{\partial d}=-\frac{1}{2}\varepsilon^{p}:\frac{\partial D^{b}}{\partial d}:\varepsilon^{p}$$

Inspired by the work of Pensee et al. [47], a liner damage criterion is considered
(40)$$f^{d}\left(Y,d\right)=Y-R\left(d\right)\leq 0$$
where *R*(*d*) is the damage energy release threshold at a given value of the damage. Based on the previous work by Zhu and Shao [45], the rock strength is attained when *R*(*d*) takes its maximum value at $d=d_{c}$. $d_{c}$ is the microscopic damage variable at the peak stress. Also, the model should reflect the strain hardening behavior when $d\leq d_{c}$ while the model should predict the damage softening behavior when *d* > *d*_c_. The following power form is adopted for *R*(*d*)
(41)$$R\left(d\right)=R\left(d_{c}\right)\frac{2\xi }{1+\xi^{2}}$$
where $R\left(d_{c}\right)$ is the maximum value of *R*(*d*) at $d=d_{c}$. The ratio ξ is defined by $\xi =d/d_{c}$. In this study, the associated damage potential is adopted which means that the damage potential is equal to the damage criterion, namely $g^{d}=f^{d}$.

## 5. Elastoplastic Damage Coupling Correction Algorithm

### 5.1. Integration by Return Mapping Algorithm

Numerical implementations of the inelastic constitutive model require the stress state to be corrected and returned onto the yield surface. Generally, there are two types of numerical integration techniques: explicit algorithms and implicit algorithms. The explicit algorithms parameter updates at the beginning of a given time step. The disadvantage is that when the time step is decreased, a non-convergent and infinite loop may happen during iterations [48]. The return mapping algorithm, first proposed by Simo and Ortiz [49], is a typical implicit algorithm and is widely used for the numerical implementation of elastoplastic models in the programming of FEM. This method can reach an asymptotic global quadratic convergence rate when using the full Newton–Raphson iteration method [50].

In this study, this algorithm is extended to solve the coupled elastoplastic damage problem. If the implementations are based on the strain-controlled strategy, a trial calculation is tested with a new strain increment. If the yield condition is reached, the return mapping algorithm is utilized to drive the stress status back to the yield surface. If the yield condition is not reached, the stress status is updated using elastic prediction.

For the coupled condition, one assumes that the macroscopic stress $\sigma^{k}$ and strain $\varepsilon^{k}$, as well as the internal variables ($d^{k}$
$\Gamma^{k}$ and $\beta^{k}$) at the end of the *k*th loading step, are known. If the yield condition is reached, one has $f^{p,\left(k+1\right)}\left(\sigma ,\;\varepsilon^{p},d\right)>0$ and $f^{d,\left(k+1\right)}\left(Y,\;d\right)>0$. To drive the stress status back to the yield surfaces, the inner iteration process is taken over at the (*k* + 1)th loading step. During each inner iteration process, the plastic and damage loading conditions can be approximately expressed by using first-order Taylor expansion
(42)$$\{\begin{array}{c} f_{m+1}^{p,\left(k+1\right)}=f_{m}^{p,\left(k+1\right)}+\frac{\partial f_{m}^{p,\left(k+1\right)}}{\partial d}\delta d+\frac{\partial f_{m}^{p,\left(k+1\right)}}{\partial \beta }\delta \beta +\frac{\partial f_{m}^{p,\left(k+1\right)}}{\partial \Gamma }:\delta \Gamma +\frac{\partial f_{m}^{p,\left(k+1\right)}}{\partial \sigma }:\delta \sigma \approx 0\\ f_{m+1}^{d,\left(k+1\right)}=f_{m}^{d,\left(k+1\right)}+\frac{\partial f_{m}^{d,\left(k+1\right)}}{\partial d}\delta d+\frac{\partial f_{m}^{d,\left(k+1\right)}}{\partial \beta }\delta \beta +\frac{\partial f_{m}^{d,\left(k+1\right)}}{\partial \Gamma }:\delta \Gamma \approx 0 \end{array}$$
where subscript *m* is the number of inner iterations in the recycling of (*k* + 1)th loading step. δσ, δΓ, δβ, and δd are used to represent the increments of σ, Γ, β, and *d* determined at the *m*th iteration. The symbol ‘*δ*’ is used to represent the small increment of variables between two inner iterations.
(43)$$\delta d=\delta \lambda^{d}$$
(44)$$\delta \beta =\eta_{\phi }\delta \lambda^{p}$$
(45)$$\delta \Gamma =V\delta \lambda^{p}$$
(46)$$\delta \sigma =-D^{m}:\left(V+\frac{1}{3}\eta_{\varphi }\delta \right)\delta \lambda^{P}$$
where $\delta \lambda^{d}$ and $\delta \lambda^{p}$ are the modification of $\lambda^{d}$ and $\lambda^{p}$ during the iteration from *m*th to (*m* + 1) th. Substitution of Equation (43) to Equation (46) into Equation (42) gives
(47)$$\{\begin{array}{c} f_{m}^{p,\left(k+1\right)}+\frac{\partial f_{m}^{p,\left(k+1\right)}}{\partial d}\delta \lambda^{d}+\left[\frac{\partial f_{m}^{p,\left(k+1\right)}}{\partial \beta }\eta_{\varphi }+\frac{\partial f_{m}^{p,\left(k+1\right)}}{\partial \Gamma }:V-\frac{\partial f_{m}^{p,\left(k+1\right)}}{\partial \sigma }:D^{m}:\left(V+\frac{1}{3}\eta_{\varphi }\delta \right)\right]\delta \lambda^{p}\approx 0\\ f_{m}^{d,\left(k+1\right)}+\frac{\partial f_{m}^{d,\left(k+1\right)}}{\partial d}\delta \lambda^{d}+\left(\frac{\partial f_{m}^{d,\left(k+1\right)}}{\partial \beta }\eta_{\varphi }+\frac{\partial f_{m}^{d,\left(k+1\right)}}{\partial \Gamma }:V\right)\delta \lambda^{p}\approx 0 \end{array}$$

Finally, the plastic multiplier and damage multiplier can be obtained by solving the set of linear equations composed of plastic and damage multipliers ($\delta \lambda^{p}$, $\delta \lambda^{d}$)
(48)$$\left\{\begin{array}{c} \delta \lambda^{d}\\ \delta \lambda^{p} \end{array}\right\}=\frac{1}{A_{11}A_{22}-A_{12}A_{21}}\left[\begin{array}{cc} A_{22} & -A_{21}\\ -A_{12} & A_{11} \end{array}\right]\left\{\begin{array}{c} B_{1}\\ B_{2} \end{array}\right\}$$

In Equation (48), *A*_11_, *A*_12_, *A*_21_, *A*_22_, *B*_1_, and *B*_2_ are defined by
(49)$$A_{11}=-\frac{\partial f_{m}^{p,(k+1)}}{\partial d},\;A_{12}=\frac{\partial f_{m}^{p,\left(k+1\right)}}{\partial \sigma }:D^{m}:\left(V+\frac{1}{3}\eta_{\varphi }\delta \right)-\frac{\partial f_{m}^{p,\left(k+1\right)}}{\partial \beta }\eta_{\varphi }-\frac{\partial f_{m}^{p,\left(k+1\right)}}{\partial \Gamma }:V$$
(50)$$A_{21}=-\frac{\partial f_{m}^{d,(k+1)}}{\partial d},A_{22}=-\left(\frac{\partial f_{m}^{d,\left(k+1\right)}}{\partial \beta }\eta_{\varphi }+\frac{\partial f_{m}^{d,\left(k+1\right)}}{\partial \Gamma }:V\right)$$
(51)$$B_{1}=f_{m}^{p,(k+1)},\;B_{2}=f_{m}^{d,(k+1)}$$

The plastic strain and the damage are updated
(52)$$\varepsilon_{m+1}^{p,(k+1)}=\varepsilon_{m}^{p,(k+1)}+\left(V+\frac{1}{3}\eta_{\varphi }\delta \right)\delta \lambda^{P}$$
(53)$$d_{m+1}^{k+1}=d_{m}^{k+1}+\delta \lambda^{d}$$

Finally, the stress, elastic strain, plastic strain, and damage can be exactly updated by the returning mapping algorithm.

### 5.2. Implementation Procedure

The previous numerical integration scheme applied to phenomenological model [44] is also implemented to the micromechanical based model in this work. The flowchart of the numerical algorithm is summarized in Table 3.

## 6. Numerical Simulations

In this section, the comparison between the experimental results of the sandstone and the numerical simulations data are presented. The proposed micromechanical based elastoplastic damage model has been implemented as a user-defined in a home-made c language code. At present, only a Gauss integration point inside a finite element is studied which indicates that the simulation results are independent of the element type. In the future work, the subroutine can be easily embedded into finite element software or finite difference software to analyze the safety and stability of hydropower station in excavation and operation conditions.

### 6.1. Identification of Model Parameters

As mentioned above, the phenomenological models always introduce so many parameters to account for the coupled relationship between plasticity and damage. However, the proposed coupled elastoplastic damage model in this work only contains five parameters and all the parameters can be determined from experiments in the laboratory. The feature of the developed model makes it more suitable for the engineering application. The calibration of the parameters is discussed below.

Equation (38) shows that the major principal stress $\sigma_{1}$ increases linearly with the increase of minor principal stress $\sigma_{3}$ which indicates that the friction criterion is similar to the Mohr–Coulomb criterion in the formulation. Therefore, the generalized friction coefficient $\eta_{\varphi }$ and parameter *R*(*d*_c_) can be expressed with the cohesion *c* and internal friction angle *φ* in Equation (1)
(54)$$\{\begin{array}{c} \eta_{\varphi }=\frac{2\sqrt{6}\sin\varphi }{\left(3-\sin\varphi \right)}\\ R\left(d_{c}\right)={\left[\frac{\sqrt{6}c\cos\varphi }{\left(3-\sin\varphi \right)\sqrt{\chi }}\right]}^{2} \end{array}$$

With the parameters (c and *φ*) listed in Table. 1, the determined $\eta_{\varphi }$ and *R*(*d*_c_) are 1.56 and 0.009, respectively.

The relationship between bulk modulus $k^{m}$ and the shear modulus and $\mu^{m}$ with the elastic constants *E_s_* and υ can be expressed as:(55)$$\{\begin{array}{c} k^{m}=\frac{E_{s}}{3\left(1-2\upsilon \right)}\\ \mu^{m}=\frac{E_{s}}{2\left(1+\upsilon \right)} \end{array}$$

The Young’s modulus *E_s_* and Poisson’s ratio υ of the sandstone is shown in Figure 6. For the sack of simplicity, the average values of the Young’s modulus *E_s_* = 15.86 GPa and Poisson’s ratio υ=0.198 are selected.

According to Lockner [51], the maximum damage variable *d*_c_ (refers to the crack density at peak point) can be determined from the acoustic emission tests. Unfortunately, we do not have such a device in our laboratory. To calibrate the parameter *d*_c_, parametric studies on the influence of *d*_c_ upon material responses are carried out and the results are given in Figure 11. The parameter *d*_c_ influences the shape of the stress–strain curve in the post-peak region. With the increases in *d*_c_, the ductile characteristic is more significant. Even the confining pressure increases to 40 MPa, the sandstone is with brittle failure. According to the sensitivity analysis results of *d*_c_ and the shapes of the stress–strain curve of the test sandstone, we take approximately *d*_c_ = 1.0 for the test sandstone.

Finally, the parameters of the model are listed in Table 4.

### 6.2. Simulation of Experimental Results of Test Sandstone

The results of each modeled test are compared to the experimental results of the sandstone (Figure 12). There is a good concordance between the modeled results and experimental data under different confining pressures. However, some systematic differences for the tests are also observed. The difference between the numerical curve and the experimental data under uniaxial compression stems from the compelling concave upward stage of sandstone under low confinements. The experimental data show a marked decrease of the stress in the post-peak region while a smooth fall of stress is received from the numerical model. This could be attributed to the shear localization and stress-induced fractures resulting from the coalescence near the peak stress of clusters of oriented microcracks. This behavior of sandstone is behind the topic of this study and will be developed in our ongoing works.

### 6.3. Simulation of Experimental Results of Shandong Red Sandstone

To illustrate the utility and consistency of the proposed coupled elastoplastic damage model, the mechanical properties of Shandong sandstone are here investigated. Figure 13 shows the experimental results and numerical simulations of triaxial compression tests performed on Shandong sandstone. Following the procedure presented in Section 5.1, the model’s parameters of Shandong sandstone are determined and listed in Table 5.

The comparisons show a good agreement between numerical predictions and experimental results for five levels of confining pressures, namely 5, 20, 35, 50, 65 MPa. The strain hardening in the pre-peak region and strain softening in the post-peak region are clearly produced. The dependence of mechanical properties of sandstone on confining pressure is well simulated. In conclusion, the developed model can simulate the coupled elastoplastic damage properties of sandstone at the scale of the rock sample.

## 7. Conclusions

The mechanical properties of studied sandstone in this work is significantly important to the stability and safety of the hydropower station in Southwest China. A series of uniaxial and triaxial compression tests were carried out in our laboratory. The strength and deformation characteristics were investigated. The microstructure of the failure specimen was examined through SEM. The results showed that the complete stress–strain curves of the sandstone can be divided into four stages. The volume of the sandstone changes from compaction to dilatancy during the failure process. The critical stresses all depend on the confining pressure. The Young’s modulus increases nonlinearly with the confining stress while the relationship between the Poisson’s ratio and the confining stress is not clear. The plastic strain increment $N_{\phi }$ and the dilation angle *ϕ* show a negative response to the confining pressure. The SEM images present the growth and frictional sliding of the cracks induced by the stress.

According to the experimental results, a coupled elastoplastic damage model was developed based on the irreversible thermodynamic framework. Specific plastic and damage criteria based on the micromechanics are proposed to describe the physical process of propagation and frictional sliding of microcracks. Compared with the phenomenological model, the model developed in this paper can reflect the physical mechanism of the failure of the sandstone that observed from the tests. Besides, the model only has five parameters and each one can be determined from laboratory tests. The general constitutive integrated formulations of the coupled elastoplastic damage model were deduced based on the return mapping algorithm. The model was validated through a comparison of the numerical simulation results to the experimental data. A good concordance between the modeled results and experimental data suggests that this model can provide a good representation of the nonlinear behavior of the sandstone.

In this study, we mainly present the experimental results of the sandstone, and a micromechanical based elastoplastic damage model was developed. In the near future, the proposed constitutive model will be implemented into the finite element method (FEM). Combining the laboratory tests with the in-situ tests, the mechanical parameters can be determined. Using the developed model, the excavation damaged zones (EDZs) can be estimated, and it could provide a reference for the design and construction of a huge hydropower project.

## Figures and Tables

**Figure 1 materials-13-03414-f001:**
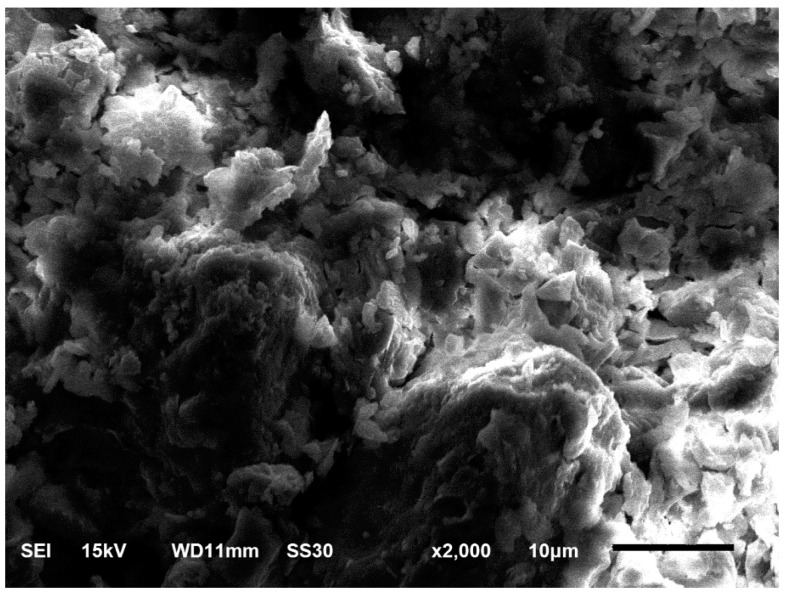
Scanning electron microscope image of tested sandstone.

**Figure 2 materials-13-03414-f002:**
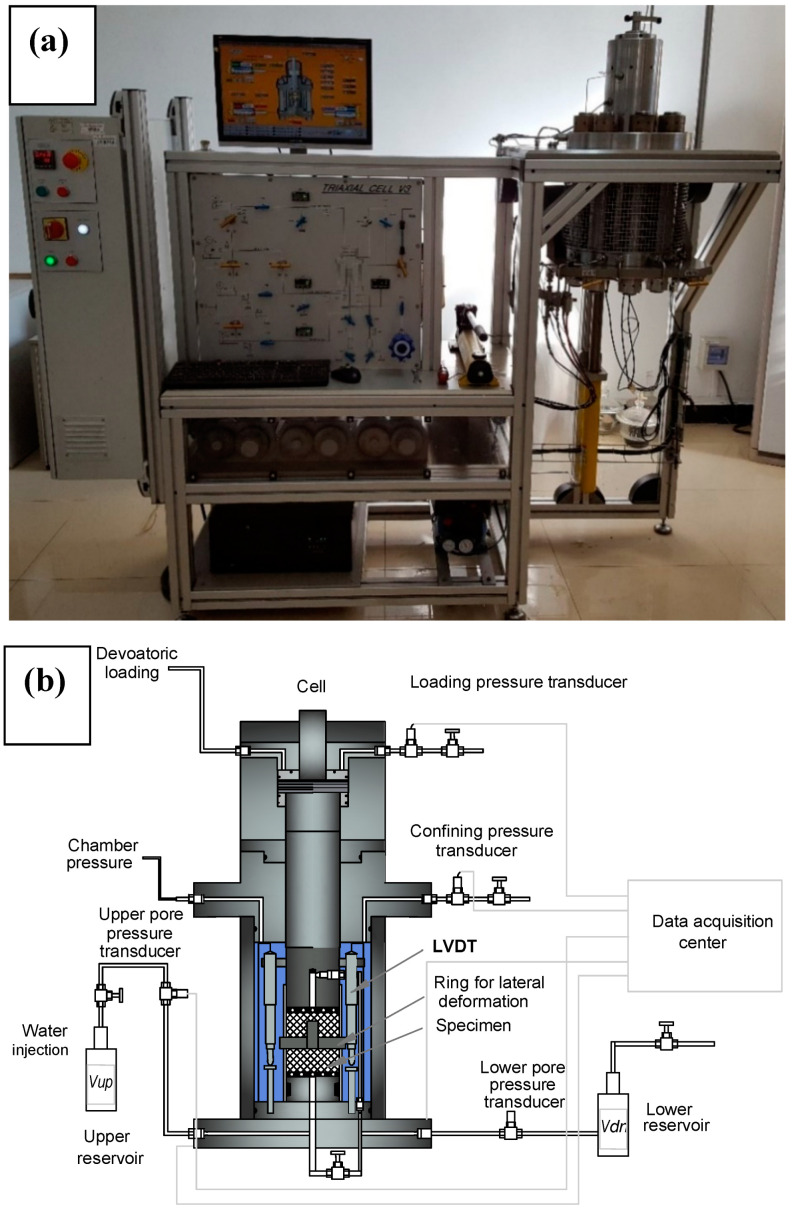
Photo (**a**) and schematic diagram (**b**) of the experimental system.

**Figure 3 materials-13-03414-f003:**
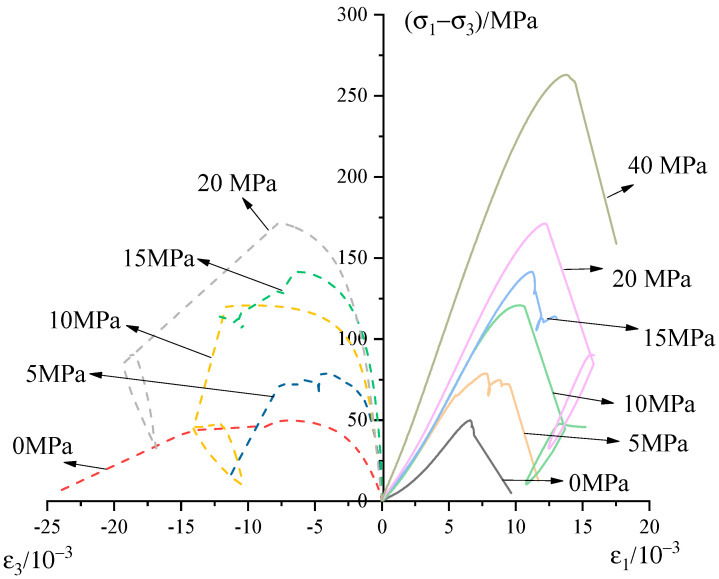
Stress–strain curves of sandstone under different confining pressures.

**Figure 4 materials-13-03414-f004:**
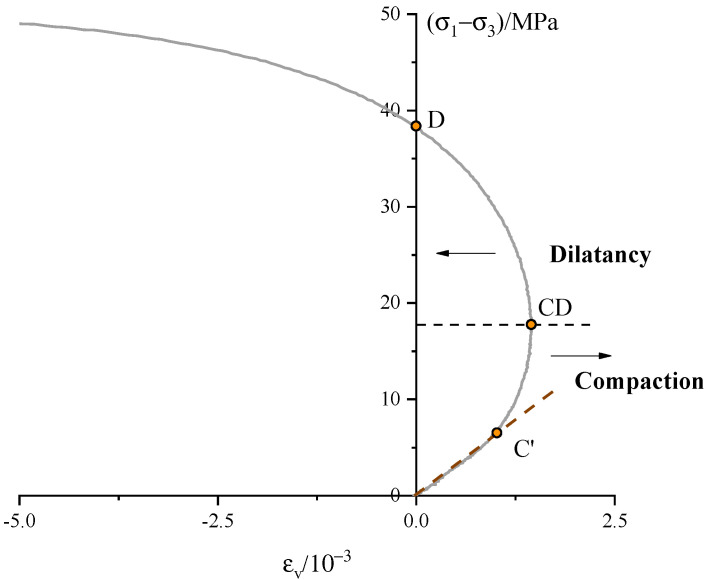
Volumetric strain versus deviatoric stress curve under uniaxial compression.

**Figure 5 materials-13-03414-f005:**
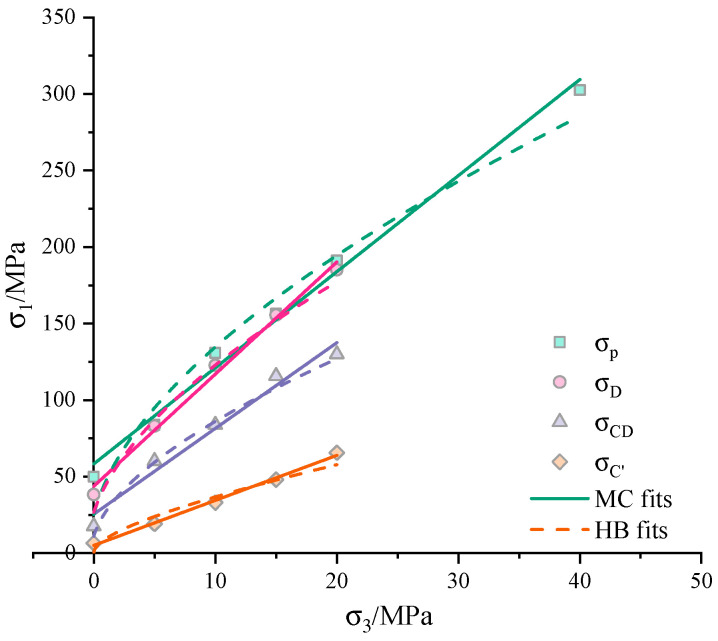
Stresses data and Mohr–Coulomb (MC) and Hoek–Brown (HB) fits for the sandstone.

**Figure 6 materials-13-03414-f006:**
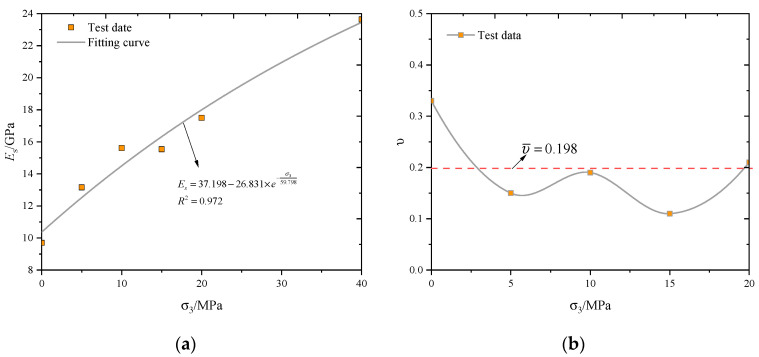
Influence of confining pressure on (**a**) Young’s modulus and (**b**) Poisson’s ratio of sandstone.

**Figure 7 materials-13-03414-f007:**
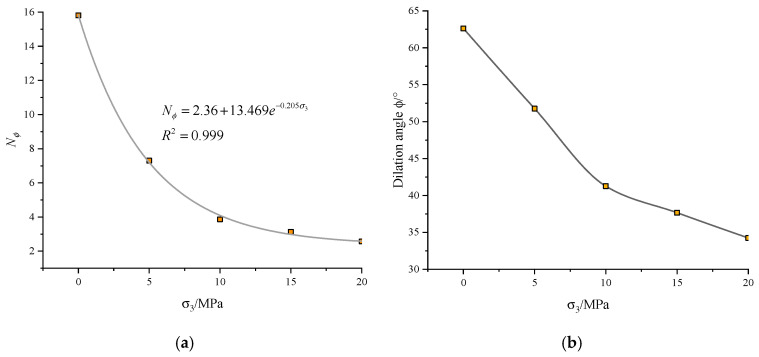
The evolution of (**a**) plastic strain increment $N_{\phi }$ and (**b**) dilation angle ϕ with the confining pressure.

**Figure 8 materials-13-03414-f008:**
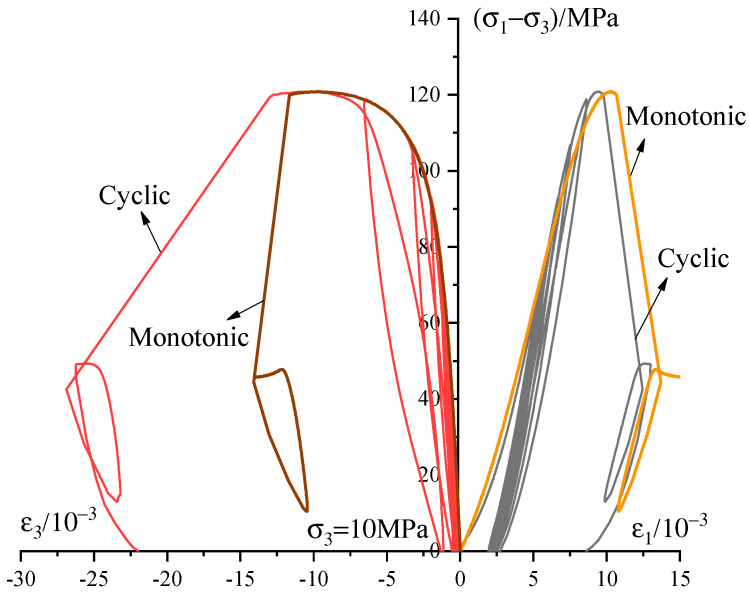
Cyclic stress–strain curve of sandstone under confining pressure of 10 MPa.

**Figure 9 materials-13-03414-f009:**
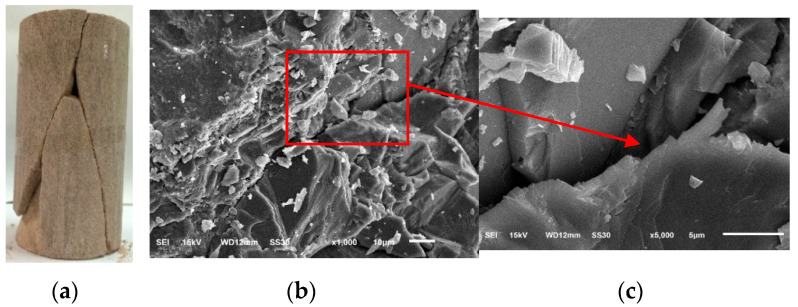
Failed sandstone specimens (**a**) and SEM images of microcracks in the specimen under uniaxial compression (**b**,**c**).

**Figure 10 materials-13-03414-f010:**
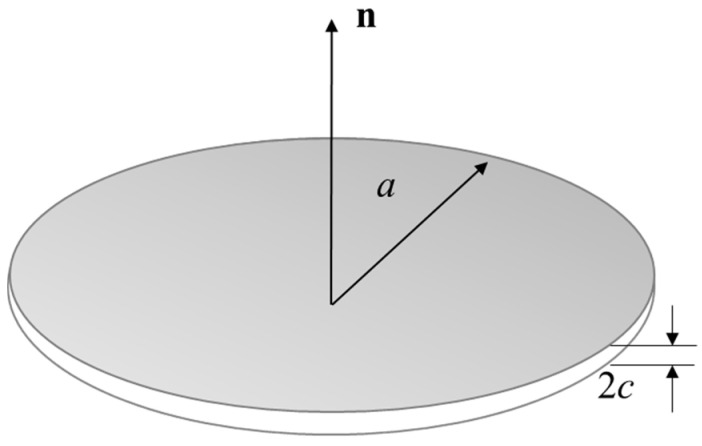
Schematic representation of a penny-shaped crack.

**Figure 11 materials-13-03414-f011:**
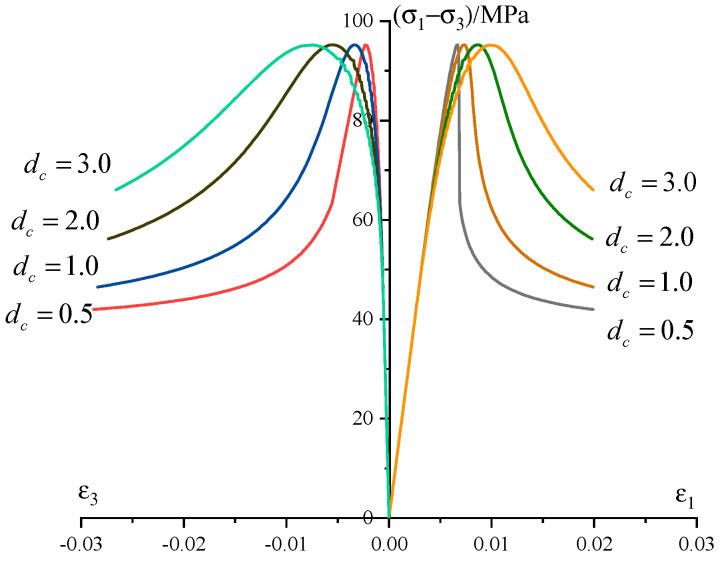
Sensitivity analysis on the parameter *d*_c._

**Figure 12 materials-13-03414-f012:**
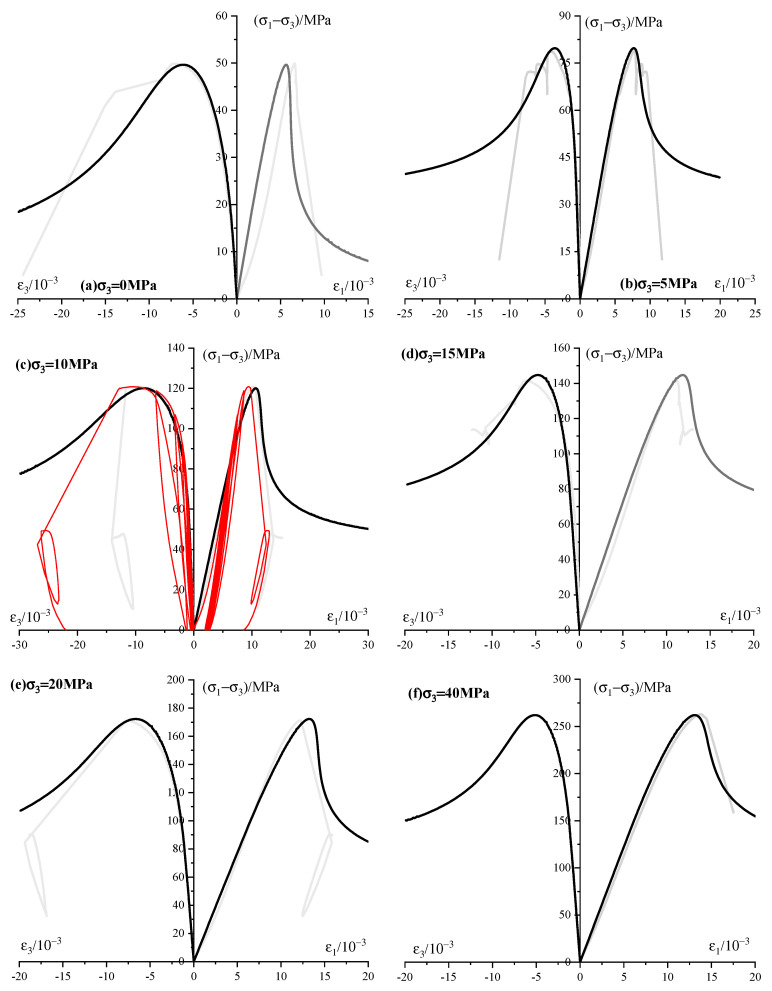
Model results (black) compared to experimental responses (grey). The cyclic experimental results are shown in red, the left is stress–radial strain curves and the right is stress–axial strain curves.

**Figure 13 materials-13-03414-f013:**
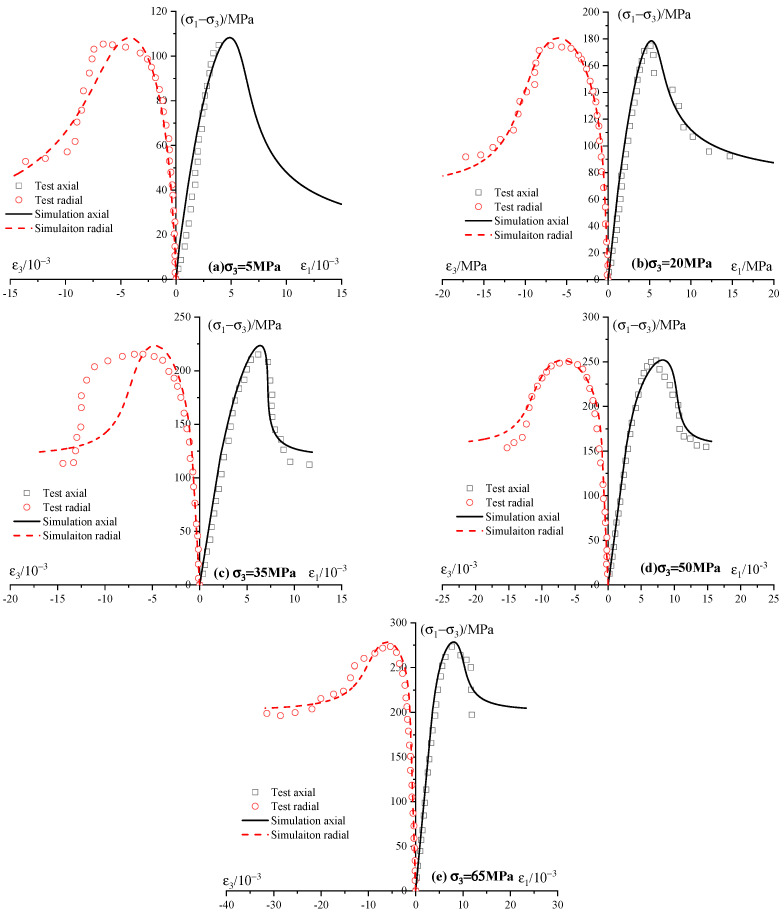
Model results compared to experimental responses of Shandong sandstone (Test data collected from [52]).

**Table 1 materials-13-03414-t001:** Critical stresses during the dilation and the peak stress of sandstone.

$\sigma_{3}/\mathrm{MPa}$	$\sigma_{C^{\prime }}/\mathrm{MPa}$	$\sigma_{CD}/\mathrm{MPa}$	$\sigma_{D}/\mathrm{MPa}$	$\sigma_{p}/\mathrm{MPa}$
0	6.558	17.658	38.259	49.933
5	14.093	55.263	78.135	78.797
10	22.967	73.777	112.724	120.904
15	33.081	100.74	140.601	141.510
20	45.604	109.955	165.122	171.250
40	-	-	-	262.597

**Table 2 materials-13-03414-t002:** Stress parameters of the sandstone using the linear Mohr–Coulomb and non-linear Hoek–Brown.

Points	UCS/MPa	*m*	*s*	*a*	φ	*c*/MPa	*R^2^*(HB)	*R^2^*(MC)
$C^{\prime }$	0.096	746.694	1.0	0.5	29.511	1.471	0.966	0.996
CD	11.102	51.033	1.0	0.5	44.190	5.380	0.974	0.974
D	26.738	44.963	1.0	0.5	49.440	8.079	0.977	0.992
peak	25.295	58.911	1.0	0.5	46.491	11.633	0.973	0.992

**Table 3 materials-13-03414-t003:** Flowchart of return mapping algorithm of the micromechanical based model.

Load to next step $\varepsilon^{k+1}=\varepsilon^{k}+\Delta \varepsilon^{k+1}$, $\sigma^{k+1}=\sigma^{k}+D^{m}:\Delta \varepsilon^{k+1}$; $d^{k+1}=d^{k}$, $\Gamma^{k+1}=\Gamma^{k}$, $\beta^{k+1}=\beta^{k}$;
Plastic correction Determine the plastic multiplier $\delta \lambda^{p}$ with Equation (47) while $\delta \lambda^{d}=0$. Update the variables: $\Gamma^{k+1}=\Gamma^{k+1}+\delta \lambda^{p}V$, $\beta^{k+1}=\beta^{k+1}+\delta \lambda^{p}\eta_{\varphi }$, $d^{k+1}=d^{k}$; Update the macroscopic stress and strain:$\{\begin{array}{c} \varepsilon^{k+1}=\Gamma^{k+1}+\frac{1}{3}\beta^{k+1}\delta \\ \sigma^{k+1}=\sigma^{k+1}-D^{m}:\left(V+\frac{1}{3}\eta_{\varphi }\delta \right)\delta \lambda^{p} \end{array}$
Elastoplastic damage coupling correction Calculate the multipliers {$\delta \lambda^{p}$, $\delta \lambda^{p}$} using the Equation (48) Update the variables: $\Gamma^{k+1}=\Gamma^{k+1}+\delta \lambda^{p}V$, $\beta^{k+1}=\beta^{k+1}+\delta \lambda^{p}\eta_{\varphi }$, $d^{k+1}=d^{k}+\delta \lambda^{d}$; Update the macroscopic stress and strain: $\{\begin{array}{c} \varepsilon^{k+1}=\Gamma^{k+1}+\frac{1}{3}\beta^{k+1}\delta \\ \sigma^{k+1}=\sigma^{k+1}-D^{m}:\left(V+\frac{1}{3}\eta_{\varphi }\delta \right)\delta \lambda^{p} \end{array}$

**Table 4 materials-13-03414-t004:** Model’s parameters for sandstone.

Parameters	*E_s_* (GPa)	υ	$\eta_{\varphi }$	$R\;(d_{c})$	$d_{c}$
Values	15.86	0.198	1.56	0.009	1.0

**Table 5 materials-13-03414-t005:** Model’s parameters for Shandong sandstone.

Parameters	*E_s_* (GPa)	υ	$\eta_{\varphi }$	$R\;(d_{c})$	$d_{c}$
Values	46.0	0.33	1.16	0.034	2.57

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
