# Peer review of "Experimental Investigation and Micromechanical Modeling of Elastoplastic Damage Behavior of Sandstone"

_materials, 2020, doi:10.3390/ma13153414_

Round 1
Reviewer 1 Report
Remark 1. Figure 3 needs some additional information, e.g., there is no description for some of the curves presented in the chart. Please correct.
Remark 2. Lines 201, 248,261, 262, 273, 276, 316 and many others, instead of “Error! Reference source not found.” please enter the correct description. The entire article needs to be checked in this regard.
Remark 3. In Figure 8, please add the description of the curves presented in the chart.
Remark 4. In line 301, the Authors use ar, but in line 304 ar was used. Is this correct?
Remark 5. Equation 22, the Authors use , but it is not described. Please add the description.
Remark 6. In line 314, the Authors nm, but it is not described. Please add the description.
Remark 7. In the part 4.3, equation numbering needs to be adjusted.
Remark 8. In Figure 11, there is no description for some of the curves presented in the chart. Same for figure 13. Please correct.
Remark 9. Please review style and format of the paper, there are many improvements that need to be made.
Reviewer 2 Report
The manuscript entitled "Experimental investigation and micromechanical modeling of elastoplastic damage behavior of sandstone” conducted a series of triaxial compression tests under different confining pressures to investigate the mechanical behavior of the sandstone collected from the dam site of a hydropower station in Southwest China. A coupled elastoplastic damage model was proposed based on the experimental results.
The manuscript is presenting an interesting topic related to a type of soil under the foundation of a dam in China. The manuscript presents valuable experimental and numerical results. The manuscript is good in quality and well written. However, a few technical comments should be addressed before acceptance.
- The authors should highlight the main conclusions derived from their study in the abstract section. The authors did not specify if the dam is under construction or design. So, how this research helped to provide guides in the design and construction of the dame under this study.
- The authors should increase their discussion on previous related research and highlight how their study is providing a different approach or adding significantly to what has been done.
- An on-site photo of the real test should be added to the revised manuscript in Figure 2.
- In section 2.2: Which standards did the authors follow to carry out the tri-axial tests.
- Line 184: It should be "higher" not "high".
- Line 201: The authors should fix this text error through the manuscript.
- Line 435: The authors should provide a flow chart describing the sequences of the proposed numerical model.
- The conclusion section should include how this research helped in the design and construction of the dam under study.
